# The Influence of Cutting Parameters on Plastic Deformation and Chip Compression during the Turning of C45 Medium Carbon Steel and 62SiMnCr4 Tool Steel

**DOI:** 10.3390/ma15020585

**Published:** 2022-01-13

**Authors:** Marcel Kuruc, Tomáš Vopát, Jozef Peterka, Martin Necpal, Vladimír Šimna, Ján Milde, František Jurina

**Affiliations:** Faculty of Materials Science and Technology, Slovak University of Technology in Bratislava, Vazovova 5, 81243 Bratislava, Slovakia; tomas.vopat@stuba.sk (T.V.); jozef.peterka@stuba.sk (J.P.); martin.necpal@stuba.sk (M.N.); vladimir.simna@stuba.sk (V.Š.); jan.milde@stuba.sk (J.M.); frantisek.jurina@stuba.sk (F.J.)

**Keywords:** plastic deformation, chip compression, cutting edge radius, cutting zone

## Abstract

The paper deals with the issue of cutting zone and chip compression. The aim was to analyse the microstructure transverse section of the cutting zone on a metallographic cut, due to determined values of chip compression and plastic deformation, which affect the cutting process efficiency. The tested cutting tool material was coated with cemented carbide. The selected workpiece materials were C45 medium carbon steel of ISO grade and 62SiMnCr4 tool steel of ISO (W.Nr. 1.2101) grade. In the experiments, a DMG CTX alpha 500 turning centre was used. The cutting speed and feed were varied, and the depth of the cut was kept constant during the turning. The plastic deformation and chip compression determine the efficiency of the cutting process. The higher compression requires more work to perform the process and, therefore, it requires more energy for doing so. With the increase of the cutting speed, the deformation for C45 steel is decreased. The rapid deformation reduction was observed when the cutting speed was increased from 145 m/min to 180 m/min. Generally, deformation is decreasing with the increase of the feed. Only at a cutting speed of 145 m/min was the deformation elevation observed, when the feed was increased from 0.4 mm to 0.6 mm. During the turning of the 62SiMnCr4 tool steel we observed an error value at a cutting speed of 145 m/min and a feed of 0.4 mm was the middle cutting parameter. However, feed dependence was clear: With an increase of the feed, the plastic deformation was decreasing. This decreasing was more rapid with the increasing of the cutting speed. Besides plastic deformation, there was analysed chip compression as well. With the increasing of the cutting speed, there was a decrease of the chip compression. Due to a lack of information in the area of the chip compression and the plastic deformation in the cutting process, we decided to investigate the cutting zone for the turning of tool steels 62SiMnCr4, which was compared with the reference steel C45. The results could be applied to increase the efficiency of the process and improvement of the surface integrity.

## 1. Introduction

In the cutting process, the cutting edge penetrates the workpiece material. The material is, therefore, deformed plastically and it is slides off along the rake face of the cutting tool. This process is called a chip formation. The behaviour of the material in the chip formation can be investigated within the orthogonal plane. It is possible because essential parts of the material flow operate within this plane. A few methods have been created to visualize the process of deformation in front of the cutting edge. Other methods have been created to analyse the process of chip formation and the behaviour of the material in the working zone. The obtained analyses provided information about the chip formation mechanisms, the shear plane position, and the plastic deformations in the zone of chip formation. To interrupt the cutting, different methods were developed [1].

During machining operations, information about chip formation plays an essential role when optimizing tools for increased machining quality and the lifetime of the tool. Higher damage is induced in the workpiece towards the end of the tool’s lifetime; therefore, the machined surface quality decreases. Examination of the cutting zone is a relevant method to analyse the chip formation.

Jun et al. [2] explored the mechanism of a fragmented chip formation in a high-speed cutting (HSC) process. For this purpose, they used the continuum-based discrete element method (DEM). This method is based on the stress wave theory. It is suitable for capturing multiple cracks’ propagation under high-speed impact. The mechanism of the chip forming based on the stress wave theory was quantitatively analysed by DEM simulations against experimental data, which showed that the unloading wave reflected by the free surface of chips under high-speed conditions has a fundamental influence on chip formation.

Ortiz-de-Zarate et al. [3] proposed a new methodology to determine the friction and normal stress distribution on the rake face of the tool using Partially Restricted Contact Length Tools in the orthogonal cutting tests. The influence of feed, cutting speed, and coatings on tool-chip friction was studied. Tests were executed during the machining of the material AISI 1045. The results showed that the novel methodology can replace the more difficult-to-use and less robust split-tool method. They observed two obviously different contact zones: firstly, the sticking region, controlled by the shear flow stress of the workpiece material, and, secondly, the sliding region, where the friction coefficient is higher than 1.

Tang et al. [4] established an analytic model of the crack-tip field in the tip of the cutting tool, which is located in the chip root. It was performed in dry, hard, orthogonal turning (DHOT) of the hardened steel according to the metal cutting principle and fracture mechanics’ theory.

Some researchers substantially use the cutting parameters below the real application parameters, which are used in industry to simplify the analyses of the chip formation of fibre-reinforced plastics (FRP) [5]. The performed simplification showed that chips tend to be larger in magnitude at lower cutting speeds. It is known that embrittlement of the machined material is caused by higher cutting speeds. This is valid even in FRP.

The study performed by Thimm et al. [6] showed a newly developed experimental test setup. This setup was based on optical measurements using a coupled system containing a high-speed camera and a microscope. Those devices were in combination with a source of laser light. This developed method was used to determine the maximum shear strain rate and the chip speed. In the mentioned study, AISI 1045 steel was used as the workpiece material.

Researchers Baizeau [7,8] and Outeiro [9] used alternative approaches to investigate the strain field in orthogonal cutting generated by different rake angles. These approaches were based on the digital image correlation (DIC) algorithm. The authors showed how the method based on DIC can be used to determine the shear plane angle.

The approach through modelling and simulation can be applied as another tool to study the chip root. Finite element analyses were used in [10] for chip formation. In the analyses that were performed for different cutting speeds, they investigated with (1) the Johnson–Cook constitutive model, (2) a modified Johnson–Cook model known as the Hyperbolic Tangent (TANH) model (this model emphasizes the strain softening behaviour), and (3) a modified Johnson–Cook constitutive model (this model considers the temperature-dependent strain hardening factor). For the simulation of the orthogonal cutting process, they adopted a 2D Lagrangian finite element model. The results from the simulations were calculated forces and chip morphologies.

Modelling of the cutting processes can be applied to estimate the resultant cutting forces as well as a heat flow density and the temperature distribution in the examined cutting zones [11]. Analytical cutting models usually neglect a correlation between the temperature and the resistance characteristics of the examined material. However, this correlation is necessary for accurately describing the effect of the cutting conditions on the temperature. Numerous analytical models for the calculation of cutting temperatures were developed in the past. The experimental analyses of the temperature distribution in the workpieces, tools, and in the contact area between them were performed [12]. Analytical models of the temperature distribution were calculated in two heat emission zones. Those zones were the chip-forming area and the rake face of the tool. In each zone, two objects were in contact: the chip and the workpiece in the shear plane, and the chip and the tool at the rake face of the tool. The heat sources were considered with respect to the deformation of the material in the chip-forming area to determine the temperature on the rake face of the tool. The deformation and friction in the contact of the chip and the rake face were taken into consideration as well. From the joint effect of the heat sources, they calculated the temperature distribution along this contact [13].

Based on the theory of elastic–plastic material deformation, the authors [14] developed a strain-hardening mode in the cutting zone. They created the simulation model of a 2D orthogonal cutting process of the workpiece and the tool by applying the finite element method (FEM). They simulated the process of chip formation and the changes of cutting force in the machining process and analysed the condition of chip deformation and the cutting force. This method was more effective and efficient than the traditional methods; it provides a new way for cutting tool product development, research of the material’s cutting performance, and the metal cutting theory. To monitor chip root during machining of hybrid components, Denkena et al. [15] used in situ machining oscillating analysis.

In [16], the effect of rubber microparticles and silica nanoparticles on the chip formation mechanism and machining-induced damage in orthogonal cutting conditions was examined. Experimental evidence showed that the chip formation mechanism was affected by a series of discontinued fractures, which were occurring in front of the cutting tool. Chip formation in bulk and silica-modified polymer formed intermittent chips with substantial cracks at the machined surface level and subsurface in the chip formation zone.

The authors of [17] designed the multifunction measuring system. This system allows for simultaneous measuring of heat distribution, components of the cutting force, and deformation processes in the cutting zone during machining. Additionally, this was possible without interrupting the machining process.

Analysis of temperature in the cutting zone was performed by several researchers in the past, especially in the turning technology of Inconel 718. There was observed some differences of temperatures in the cutting zone measured by different researchers. They were caused by different cutting conditions [18]. In turning experiments, with a high cutting speed (up to 510 m/min), when a process liquid was applied, and when the ceramic cutting tool was used, the measured temperature was in the range between 900 °C and 1300 °C on the rake face of the cutting tool. On the rake face of the cutting tool, the temperature was higher, about 50 °C to 70 °C, than temperature on the flank face of the cutting tool [19]. Similar phenomena were observed with other superalloys [20]. When machining superalloys with ceramic tools, notch wear is present, which is correlated with variation of the chip formation mechanism [21]. Temperature distribution during superalloy machining can be better described using thermal modelling [22].

In drilling of the austenitic stainless steel 1Cr18Ni9Ti, it is necessary to examine the chip deformation process. It is because the material 1Cr18Ni9Ti belongs to the difficult-to-cut materials. In the study in [23], an experimental examination of the process of the chip transformation on the cutting edges using quick interruption of the drilling processes was carried out. The results showed that the chip deformation decreased with the increment of the distance to the chisel edge on the cutting edge and the feed rate and increased with the increment of the drilling speed during machining of 1Cr18Ni9Ti steel. A similar effect was observed in drilling AISI 1045 steel. However, the chip deformation in drilling AISI 1045 was lower than the deformation in drilling 1Cr18Ni9Ti.

To observe the chip morphology of titanium alloy, a metallographic microscope was used in the article [24]. A Digimizer image measurement software system was used to measure the geometric characteristic parameters of the chip.

There have been developed many methods for measuring temperature when milling. These methods include measuring of the temperature by monitoring of the close surface by an infrared (IR) thermographic camera as well as an IR thermometer as a remote sensor [25]. Other methods include utilizing a thermal model [26] or techniques that utilize measuring the heating cycle on the flank face of the cutting tool via optical fibre inserted into a small hole on the outer surface when milling [27]. It was observed that the thermal impact to the cutting tool was smaller in upmilling than in downmilling [28]. The machining process of Inconel 718 caused elevated temperature in the cutting zone due to high strength and low thermal diffusion of the examined material [29].

Scientists are interested in the chip, from the days of machining nesters, such as Merchant, Zorev, Asthakov, and others [30,31,32,33], who focused mainly on the description of chip formation by experimental and mathematical approaches, to the present, where attention is paid not only to mathematical methods and experimental approaches to cutting zone research but also to modelling and simulations of the resulting chip and chip deformations. A comprehensive approach to the evaluation of not only the cutting zone but also several other cutting process parameters such as cutting forces, wear, durability and service life, roughness, and quality of the machined surface is often seen, using state-of-the-art approaches, procedures, instruments, and software [34,35,36,37].

In addition to turning, milling, and drilling, cutting zone research continues for a variety of other materials [38,39] and cutting technologies such as grinding [40], milling with a disc cutter [41], and Micro and Macro Machining (MMM) [42,43] by using FEM [44] and fuzzy algorithms [45].

Pimenov and Guzeev [46] developed a model, which simulates the creation of the tension condition on the flank face of the cutting tool in orthogonal cutting. They used FEM to establish the value of the stress on the flank face of the cutting tool for various cutting modes. They developed the equation for stress intensity calculation for the flank wear of the tool, and their numerical model allowed them to determine the cutting forces for any material, any cutting mode, and flank wear and side wear.

FEM analysis was performed also by Necpal and Martinkovič [47] to investigate the mechanism of steel CK45 chip formation for the process of the orthogonal turning. By measuring the deformation of the metallographic cut, a local strain in the cutting zone was calculated. For this purpose, stereological evaluation of the grain boundary orientation was used. They focused on stress, temperature, and tool wear.

Afrasiabi et al. [48] applied the FEM (mesh-dependent) and Smoothed Particle Hydrodynamics (SPH) (mesh-free) methods to simulate the chip creation process in a thermo-mechanically coupled framework for AISI 1045 steel and Ti6Al4V titanium alloy materials. They developed a new method to measure the temperature on the rake face without necessary manipulation of the chip flow.

Knowledge of chip formation, chip deformation, chip compression and other factors offers us the possibilities of use in the practical level in optimizing the parameters of the cutting process for demanding chip technologies such as deep drilling [49], machining of composite materials [50], and machining with a hemispherical milling cutter [51,52] or in the theoretical plane for calculating and determining the effect on the roughness of the machined surface [53].

Research by other authors did not describe the method for obtaining the cutting zone. Therefore, this article shows the sample and method for obtaining the cutting zone. In addition, the article demonstrates the influence of cutting speed and feed rate on plastic deformation and chip compression during machining.

Most of the researchers were focused on temperature during the cutting process. Only a few of them dealt with the chip compression and the plastic deformation in the cutting zone, despite their importance in terms of energy efficiency and surface integrity. The chip compression and the plastic deformation are affected by the cutting conditions, as well as the machining method and the workpiece material. Due to the lack of information in this area, the purpose of this research work was to investigate the cutting zone for one of the most common machining methods (turning) and for one of the most common tool steels (62SiMnCr4). For better understanding, this tool steel was compared with the reference steel (C45). We were expecting the decrease of the chip compression and plastic deformation with the increase of cutting speed and feed rate.

## 2. Materials and Methods

### 2.1. Preparation of Material Samples and Cutting Zone Specimens

Figure 1 shows material samples that were made for obtaining the cutting zone specimens. A schematic illustration of the process of obtaining the cutting zone specimen is shown in Figure 1a [54]. Steel rods with a diameter of 4 mm were pressed into the holes, which also had a diameter of 4 mm. Thin sheets were pressed into the grooves, as seen in Figure 1b. The material sample was clamped into the hydraulic three-jaw chuck (Figure 1c). It was confirmed by experiments that the process of obtaining the cutting zone specimens for observation of the cutting zone is very reliable.

Obtained cutting zone specimens were firstly cleaned by acetone (Figure 2b), after which they were poured into the mixture of epoxy resin and heat-pressure was used on the BUEHLER SimpliMet 1000 device (BUEHLER, Leinfelden-Echterdingen, Germany). Cutting zone specimens were subsequently grinded on the BUEHLER AutoMet 3 device. After grinding, they were polished by a polishing medium of monocrystalline diamond. Then, the cutting zone specimens were cleaned and etched in a 3% nitric acid alcohol solution. Finally, metallographic cutting zone specimens (Figure 2a) were observed in the transverse section on the NEOPHOT 32 light microscope (ZEISS, Oberkochen, Germany).

### 2.2. Tested Workpiece Materials and Cutting Tool

The CNMG 120408-M6 turning inserts and PCLNL 2020K12 tool-holder by Seco were selected for this research. This cutting tool had the following geometrical parameters: a chamfer width of 0.25 mm, cutting edge length of 12 mm, rake angle of γ_o1_ = −2° (up to 0.25 mm of uncut chip thickness), rake angle of γ_o2_ = 13° (from 0.25 mm of uncut chip thickness), clearance angle of α_o_ = 6°, inclination angle of −6°, and cutting edge radius of 55 µm, as shown Figure 3.

The tested cutting tool material was coated with cemented carbide. The specific grade of the cemented carbide was TP3500 by a tool producer and the turning carbide inserts were coated by Ti (C, N) + Al_2_O_3_ coating. The selected workpiece materials were C45 medium carbon steel of ISO (AISI 1045, W.Nr. 1.050) grade as standard material and 62SiMnCr4 tool steel of ISO (STN 19 452, W.Nr. 1.2101) grade in this paper. There were no publications that investigated plastic deformation and chip compression for tool steel. The C45 medium carbon steel was normalized and annealed with the hardness max. of 225 HB. The 62SiMnCr4 tool steel was soft-annealed with the hardness max. of 225 HB. Chemical compositions of both materials are shown in Table 1 and Table 2. The authors of this article decided to research C45 steel. This steel is known as the so-called standard in determining machinability but also for other characteristics of the cutting process such as cutting forces and cutting temperature, e.g., in [55]. In addition, there is not only experimental work on this steel but also work in the field of modelling and simulation of the cutting process for this steel, e.g., in [39] analysis was made for this steel. In this article, the flow stress behaviour of C45E steel was modelled by modifying the Johnson–Cook model that incorporates the dynamic strain aging (DSA) influence.

The 62SiMnCr4 is a steel used for cold cutting tools, cutting by hand, for crushing and grinding, for clamping tools and moulds due to its excellent toughness, for hot formability, and good machinability, as well as having good resistance to impacts and wear and very good flexibility; it is utilized for scissors, cutting tools, screwdrivers, wrenches, pliers, moulds, and pins for moulding plastics. Another reason why the authors chose this material is because this steel is the subject of research awarded by the National Agency APVV and VEGA, mentioned in the acknowledgement.

### 2.3. Cutting Parameters and Chip-Forming Test

Cutting parameters that most affect the chip forming during the machining process are feed *f* and depth of cut *a_p_*. Depth of cut is related to wall thickness of samples; therefore, it was set to 3 mm. Chip-forming tests were carried out before starting the experiment because it was necessary to obtain the area of the cutting parameters where the chip forms and breaks. It was confirmed by experiment that if the cutting parameters are used in a range of feed from 0.2 to 0.6 mm and depth of cut of 3 mm, the chip will form and break (Figure 4 and Figure 5).

In the experiments, a DMG CTX alpha 500 turning centre was used. The second level of cutting speed was established for turning the C45 steel (*v_c_* = 145 m/min) and 62SiMnCr4 tool steel (*v_c_* = 110 m/min) on the basis of the recommendation from the Seco tool producer. The range of feed was selected to semi-finishing applications (*f* = 0.2 mm), medium-rouging (*f* = 0.4 mm), and roughing (*f* = 0.6 mm), with respect to an interrupted cut. Summarized cutting parameters used in experiment are shown in Table 3.

### 2.4. Cutting Zone Observation

The chip compression and the plastic deformation in the cutting zone were analysed during the turning. The obtained microstructure was observed in the transverse section of the cutting zone. A light microscope with 500× magnification was used on the metallographic cut. The structure anisotropy increased due to the plastic deformation: Grain boundaries’ orientation of the undeformed areas of the workpiece was observed. Local plastic deformation on the surface and in the cutting zone of the turned C45 workpiece can be seen in Figure 6 [56]. The local strain in the analysed area of the sample was achieved by stereological measurement of the degree of grain boundaries’ orientation [57].

Using a quantitative metallography (more specifically, Saltykov stereology methods with oriented test lines), the anisotropic microstructure was decomposed into isotropic- and planar-oriented components. The test lines were placed on a metallographic cut. The lines were parallel and perpendicular to the grain boundaries’ orientation affected by straining: In the primary area, the lines were parallel to the orientation of the texture, the slip plane (pitch angle *ϕ*_1_ in Figure 5); in the secondary area, the lines were parallel to chip/tool interface, the tool face; in the tertiary area, the lines were parallel to the machined surface. From the specific number (length unit number) of parallel test lines’ intersections with grain boundaries (*P_L_*)*_P_* and perpendicular lines’ intersections (*P_L_*)*_O_*, the planar-oriented part of the specific surface area (*S_V_*)*_OR_* of the grains and the total specific surface area (*S_V_*)*_TOT_* of the grains were calculated. The degree of orientation of the grain boundaries *O* was calculated as the ratio of (*S_V_*)*_OR_* and (*S_V_*)*_TOT_* from these values. This procedure of measurement is shown in Figure 7.

The result was proportional to the grain boundaries’ deformation degree, and the local plastic deformation was estimated by Equation (1) [56].
(1)$$\varphi =ln{\left(\frac{1+O\sqrt{2-O^{2}}}{1-O^{2}}\right)}^{\frac{2}{3}}$$
where *φ* is the local plastic deformation (-) and *O* is the degree of the grain boundaries’ orientation (°).

The cutting tool had a geometrical parameter of an end cutting angle (*α_n_*) of 6° and lip angle (*β_n_*) of 86°. The shear plane angle *ϕ* was measured directly in the microstructure specimen (Figure 6). By using these parameters, the chip compression was estimated by Equation (2) [56] for each specimen. Deformations in the cutting zone and chip compressions of each parameter are in Table 4, where deformation and chip compression were calculated according to Equations (1) and (2), respectively. The measurements were repeated three times due to statistical validity; however, only the average values are recorded in Table 4. Figure 8 and Figure 9 show the selected resultant microstructure with a shear plane angle for both materials.
(2)$$\xi =\frac{\sin\left(\phi +\left(\alpha_{n}+\beta_{n}\right)\right)}{\sin\phi }$$
where *ξ* is the chip compression (-), *ϕ* is the shear plane angle (°), *α_n_* is the end cutting angle (°), and *β_n_* is the lip angle (°).

## 3. Results and Discussion

Plastic deformation in the cutting zone was affected by the cutting parameters and the workpiece material. With the increase of the cutting speed, the deformation for C45 steel decreased. The rapid deformation reduction was observed when the cutting speed was increased from 145 m/min to 180 m/min. In the first range of the cutting speed increasing (from 110 m/min to 145 m/min) the deformation mostly slightly decreased; however, when the feed of 0.6 mm was used, the deformation was increasing. Generally, deformation was decreasing with the increasing of the feed as well. Only at the cutting speed of 145 m/min was the deformation elevation observed, when the feed was increased from 0.4 mm to 0.6 mm. The results of plastic deformation for construction steel C45 workpiece are shown in Figure 10.

During the turning of tool steel 62SiMnCr4, we observed an error value at a cutting speed of 145 m/min and feed of 0.4 mm (middle parameters). Despite that, we concluded that cutting speed does not have a clear influence on plastic deformation. At the lowest feed (0.2 mm), it was slightly increasing with the elevation of the cutting speed; but at the highest feed (0.6 mm), it was slightly decreasing with the elevation of cutting speed. However, feed dependence is clear: With the increasing of the feed, the plastic deformation was decreasing. Additionally, this decrease was more rapid with the increase of the cutting speed. The results of plastic deformation for the tool steel 62SiMnCr4 workpiece are shown in Figure 11.

Besides plastic deformation, there was analysed chip compression as well. This parameter was affected by cutting parameters and workpiece material as well; however, the behaviour of different steel materials was similar. With the increase of the cutting speed, the chip compression was decreased. For the highest feed (0.6 mm), we observed only a slight reduction. For the remaining two feeds (0.2 mm and 0.4 mm), we observed the most rapid reduction during the elevation of the cutting speed from 110 m/min to 145 m/min. With the increase of the feed, the chip formation decreased as well. The most rapid reduction was observed at the lowest cutting speed (110 m/min) when the feed was increased from 0.4 mm to 0.6 mm. The results of the chip compression for the C45 steel workpiece are shown in Figure 12.

During the turning of tool steel 62SiMnCr4, we observed similar dependences of the chip formation as during the turning of construction steel C45. With the increase of the cutting speed, the chip compression decreased. For the highest feed (0.6 mm), we observed only a slight reduction in the cutting speed elevation from 110 m/min to 145 m/min. With the increase of the feed, the chip formation decreased as well. The most rapid reduction was observed at the lowest cutting speed (110 m/min), when the feed was increased from 0.4 mm to 0.6 mm. The results of the chip compression for the 62SiMnCr4 steel workpiece are shown in Figure 13.

According to Slusarczyk [58], the tube wall thickness reduction involved two series of orthogonal turning, requiring the proper experimental studies. The first series of experiments were made for the tube wall thickness of *a_p_* = 0.5 mm. There, the chip was formed by the area on the rake face of the cutting insert. The second series of experiments were made for the tube wall thickness of *a_p_* = 1.77 mm. The components of the cutting force, such as *F_c_* (tangential component) and *F_f_* (feed component), were defined experimentally during each experimental series. The authors found that the cutting force components *F_c_* and *F_f_* were slightly decreased when the cutting speed was increased, which is the same phenomenon as chip compression.

Decreasing the character of chip compression with the increasing of the cutting speed confirmed the findings of Shalaby and Veldhuis [59] in their research of high-speed machining of hardened steel AISI 4340, where they used cutting speeds in a range from 150 to 1000 m/min and they observed a half value of the chip compression ratio at the highest cutting speed in comparison with the lowest cutting speed.

Compression deformations were studied by Li et al. [60] as well. They investigated compression deformation of the titanium alloy Ti-6Al-4V during high-speed cutting. They used cutting speeds in the range from 10 to 160 m/min and they observed that increasing the cutting speed caused an increase in the cutting temperature, thrust force, shear angle, compression value, strain, and strain rate, decreasing the cutting force and compression stress.

According to those and other researchers, we concluded that the achieved behaviour of chip compression and plastic deformation on cutting speed and feed rate observed on C45 and 62SiMnCr4 steel materials can be expected even on other metals and alloys.

## 4. Conclusions

The plastic deformation and chip compression determine the efficiency of the cutting process. The higher compression requires more work to perform the process, and, therefore, it requires more energy to do so. This energy limits the power of the machine tool and the cutting process itself: Energy used for deformation and compression at improper cutting parameters should be used to increase productivity (i.e., material removal rate) at proper cutting parameters without using a machine tool with a higher power. That is the reason why those values should be as low as possible. In the performed experiments their values were in the range of:0.21–0.70 of plastic deformation for C45 steel;0.42–0.45 of plastic deformation for 62SiMnCr4 steel;1.00–2.01 of chip compression for C45 steel; and1.06–2.46 of chip compression for 62SiMnCr4 steel.

We obtained a 0.78 plastic deformation for 62SiMnCr4; however, it was considered as an error value because other values were in a much narrower interval. This error could have been caused during the improper manipulation of the brittle cutting zone or during metallography preparation procedures.

According to the performed experiments, we generally concluded that plastic deformation as well as chip compression were decreasing with the increase of the cutting speed and feed rate. This was confirmed for both steel materials: construction steel C45 and tool steed 62SiMnCr4. According to the obtained results, we can recommended the following cutting parameters:for turning of the C45 medium carbon steel at a depth of cut of 3 mm: a cutting speed of 180 m/min and a feed of 0.6 mm;for turning of the 62SiMnCr4 steel at a depth of cut of 3 mm: a cutting speed of 145 m/min and a feed of 0.6 mm;

In those parameters, we obtained plastic deformation of 0.21 and chip compression of 1.00 for the C45 steel material and plastic deformation of 0.42 and chip compression of 1.06 for the 62SiMnCr4 steel material.

The obtained results should help to understand the cutting process. Additionally, they could be applied to increase the efficiency of the process and improve the surface integrity by selecting those values of the cutting parameters that allow the lowest chip compression and the plastic deformation in the cutting zone.

In further work, the residual stresses in the surface layer as well as the resulting surface roughness will be investigated. We also plan to expand the experiments by the finite element method analysis.

## Figures and Tables

**Figure 1 materials-15-00585-f001:**
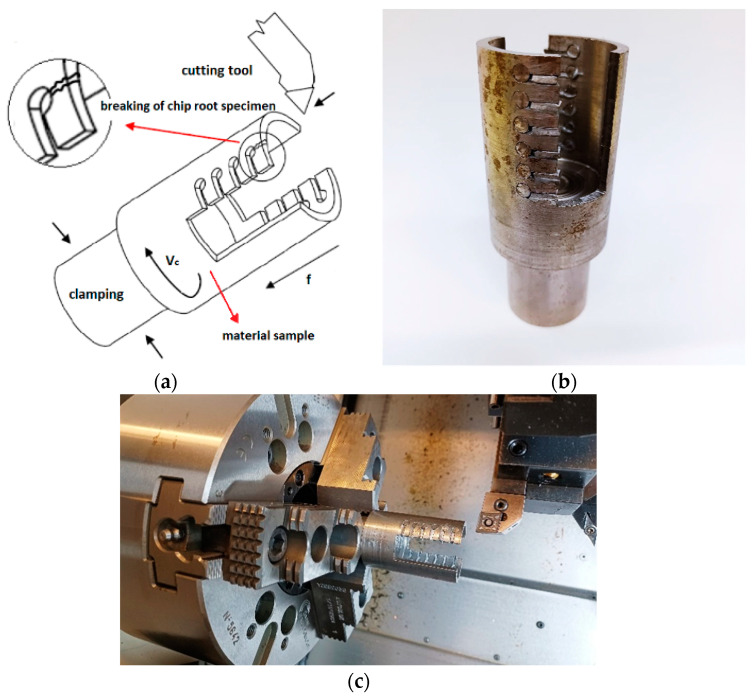
Material samples for obtaining the cutting zone. (**a**) Schematic illustration of the cutting zone, obtained from [54], (**b**) the Cutting zone samples, and (**c**) the Process of obtaining the cutting zone.

**Figure 2 materials-15-00585-f002:**
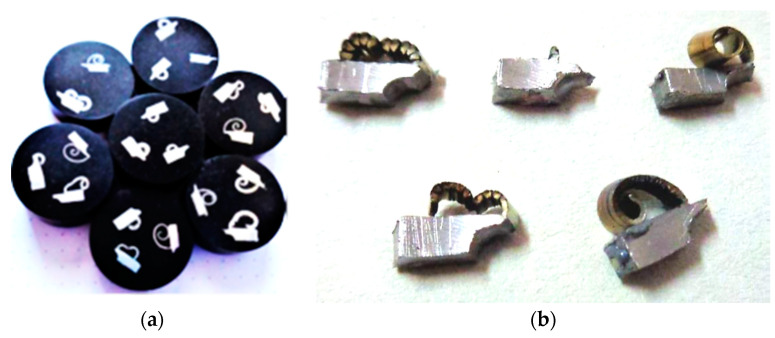
Cutting zone. (**a**) Metallographic cutting zone specimens; (**b**) Obtained cutting zone specimens.

**Figure 3 materials-15-00585-f003:**
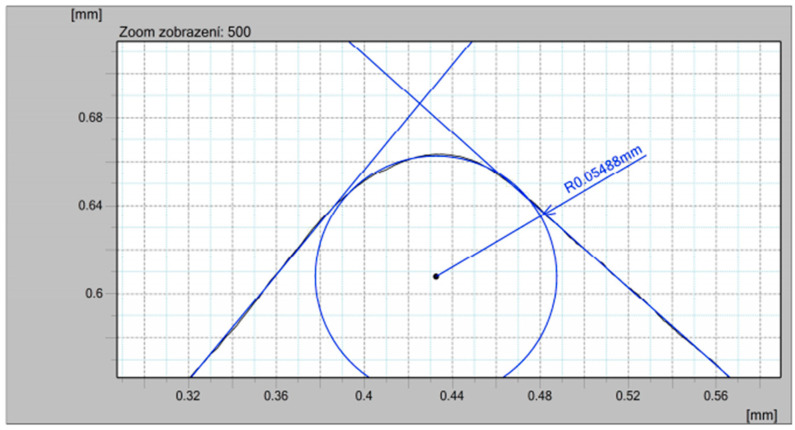
Cutting edge radius obtained by contour graph.

**Figure 4 materials-15-00585-f004:**
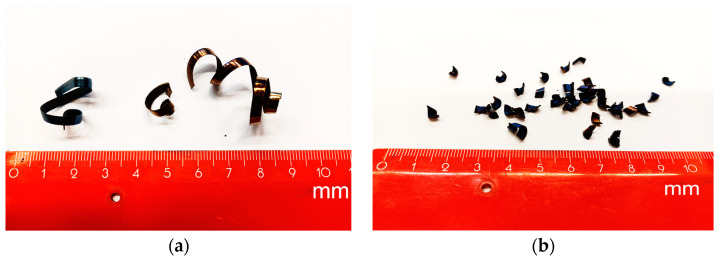
Chips after machining C45 medium carbon steel when *v_c_* = 180 m/min and feed: (**a**) *f* = 0.2 mm; (**b**) *f* = 0.4 mm; (**c**) *f* = 0.6 mm.

**Figure 5 materials-15-00585-f005:**
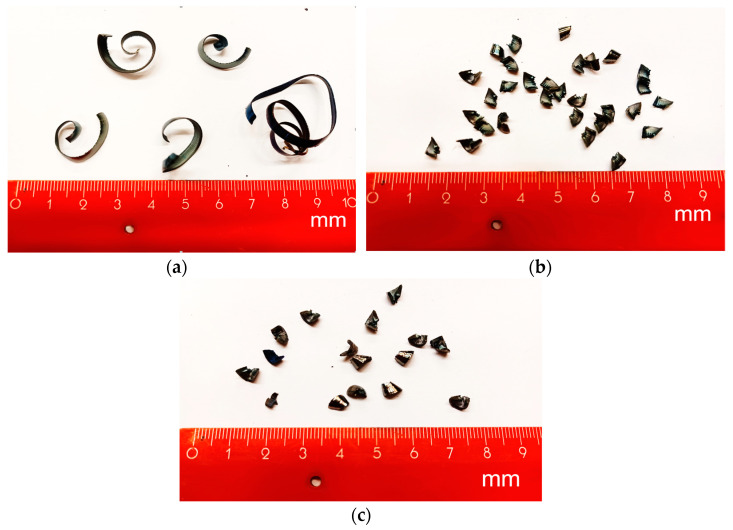
Chips after machining 62SiMnCr4 tool steel, *v_c_* = 145 m/min and feed: (**a**) *f* = 0.2 mm; (**b**) *f* = 0.4 mm; (**c**) *f* = 0.6 mm.

**Figure 6 materials-15-00585-f006:**
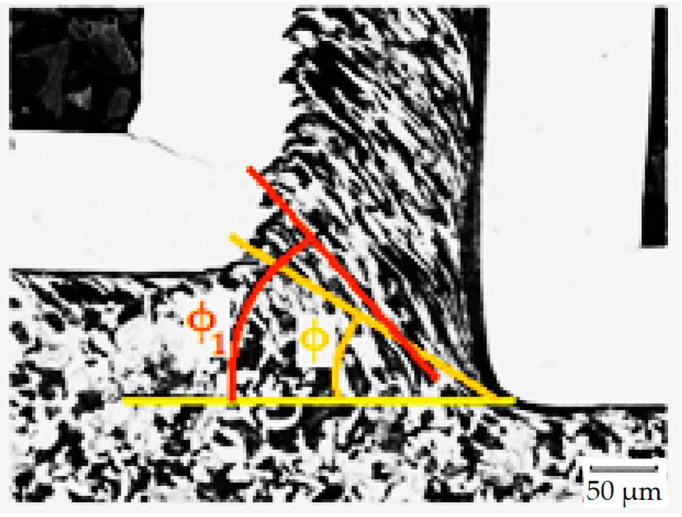
Local plastic deformation in the chip and in the cutting zone of a turned C45 workpiece.

**Figure 7 materials-15-00585-f007:**
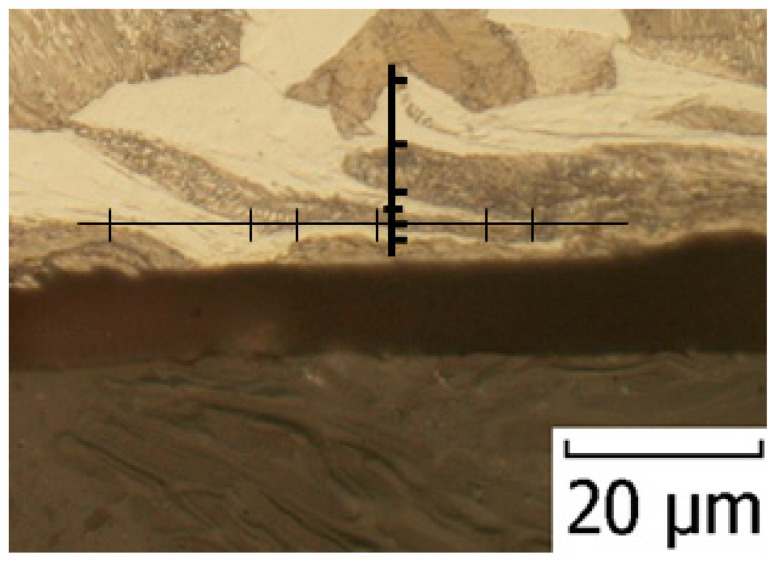
Test lines Placed perpendicular and parallel to the grain boundaries.

**Figure 8 materials-15-00585-f008:**
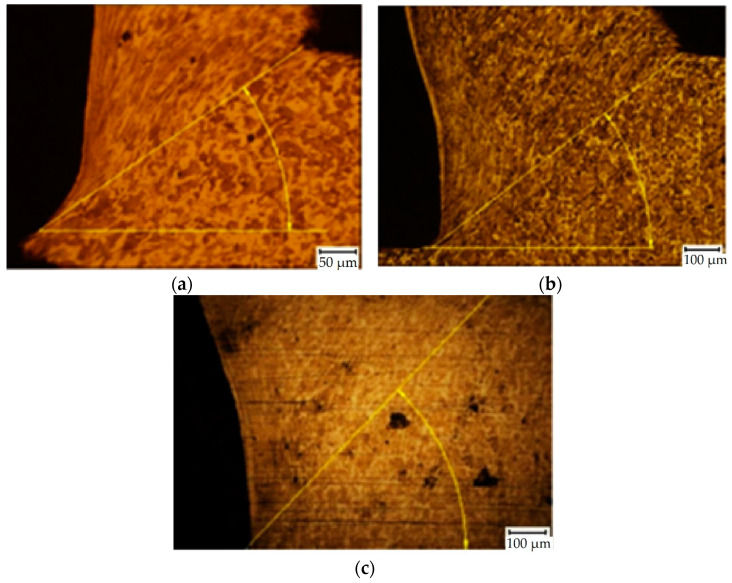
Examples of SEM micrographs of the cutting zone (C45 medium carbon steel, *v_c_* = 180 m/min): (**a**) specimen no. 7 (*f* = 0.2 mm), *ϕ* = 37°; (**b**) specimen no. 8 (*f* = 0.4 mm), *ϕ* = 39°; (**c**) specimen no. 9 (*f* = 0.6 mm), *ϕ* = 49°.

**Figure 9 materials-15-00585-f009:**
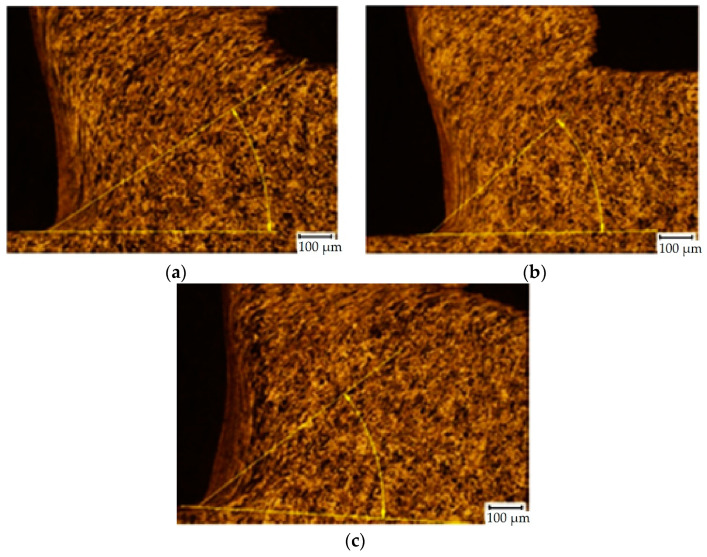
Examples of SEM micrographs of the cutting zone (62SiMnCr4 tool steel, *v_c_* = 145 m/min): (**a**) specimen no. 16 (*f* = 0.2 mm), *ϕ* = 36°; (**b**) specimen no. 17 (*f* = 0.4 mm), *ϕ* = 43°; (**c**) specimen no. 18 (*f* = 0.6 mm), *ϕ* = 44°.

**Figure 10 materials-15-00585-f010:**
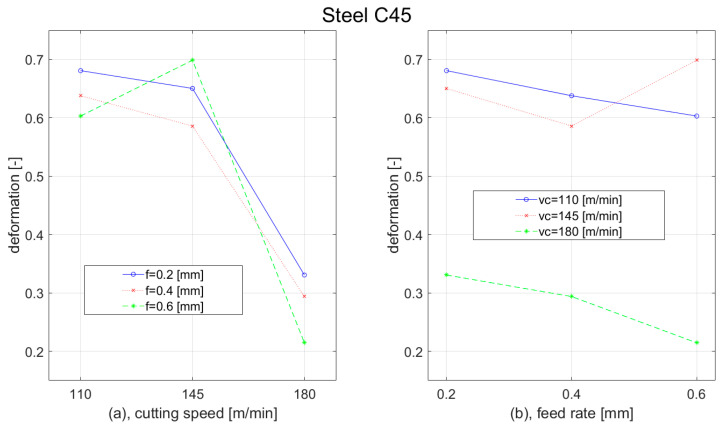
Graph of deformation dependence of Steel C45 on: (**a**) cutting speed; (**b**) feed rate.

**Figure 11 materials-15-00585-f011:**
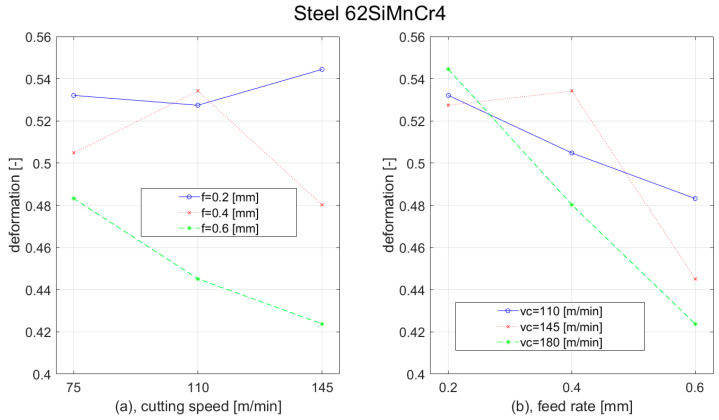
Graph of deformation dependence of Steel 62SiMnCr4 on: (**a**) cutting speed; (**b**) feed rate.

**Figure 12 materials-15-00585-f012:**
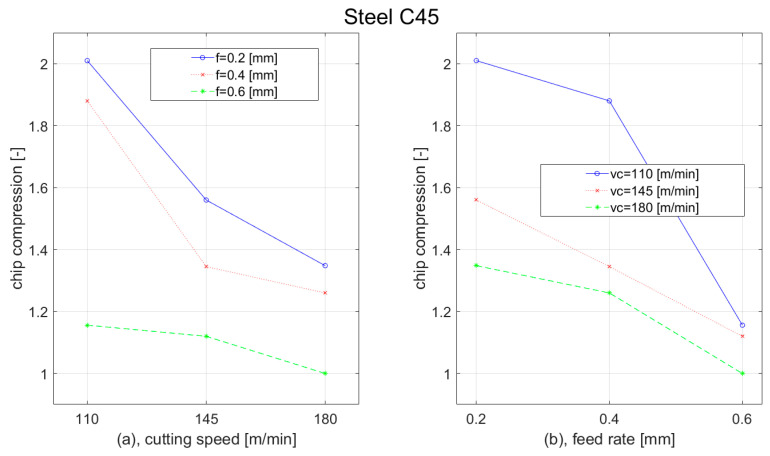
Graph of chip compression dependence of Steel C45 on: (**a**) cutting speed; (**b**) feed rate.

**Figure 13 materials-15-00585-f013:**
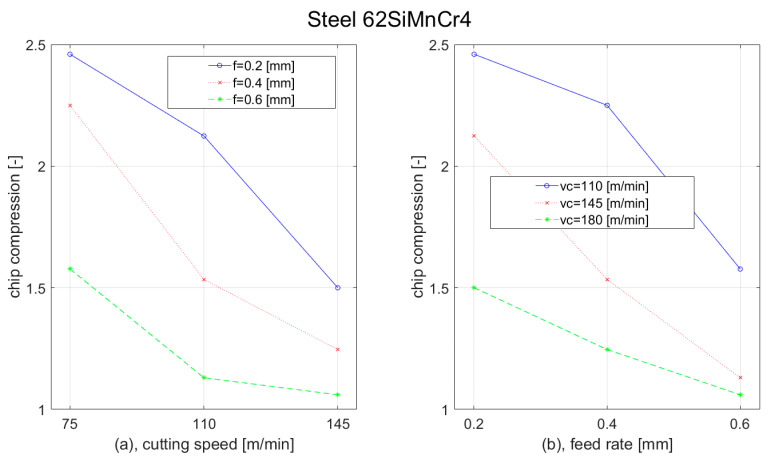
Graph of chip compression dependence of Steel 62SiMnCr4 on: (**a**) cutting speed; (**b**) feed rate.

**Table 1 materials-15-00585-t001:** Chemical composition of the C45 machined material (wt.%).

C	Si	Mn	P	S	Cr	Ni	Mo
0.43–0.50	max 0.4	0.5–0.8	max 0.045	max 0.045	max 0.4	max 0.4	max 0.1

**Table 2 materials-15-00585-t002:** Chemical composition of the 62SiMnCr4 machined material (wt.%).

C	Si	Mn	P	S	Cr
0.58–0.66	0.9–1.2	0.9–1.2	max. 0.03	max. 0.03	0.4–0.7

**Table 3 materials-15-00585-t003:** Cutting parameters used in experiment.

C45 Medium Carbon Steel				62SiMnCr4 Tool Steel			
Spec. No.	*v_c_* (m/min)	*f* (mm)	*a_p_* (mm)	Spec. No.	*v_c_* (m/min)	*f* (mm)	*a_p_* (mm)
1	110	0.2	3	10	75	0.2	3
2		0.4		11		0.4	
3		0.6		12		0.6	
4	145	0.2		13	110	0.2	
5		0.4		14		0.4	
6		0.6		15		0.6	
7	180	0.2		16	145	0.2	
8		0.4		17		0.4	
9		0.6		18		0.6	

**Table 4 materials-15-00585-t004:** Cutting parameters used in the experiment.

C45 Medium Carbon Steel			62SiMnCr4 Tool Steel		
Spec. No.	Deformation	Chip Compression	Spec. No.	Deformation	Chip Compression
1	0.680685	2.010	10	0.532105	2.460
2	0.637729	1.880	11	0.504818	2.250
3	0.602918	1.156	12	0.483215	1.577
4	0.650248	1.560	13	0.527434	2.124
5	0.585835	1.345	14	0.534136	1.534
6	0.698620	1.120	15	0.445155	1.130
7	0.330676	1.348	16	0.544442	1.500
8	0.293802	1.260	17	0.480216	1.246
9	0.214603	1.000	18	0.423815	1.060

## Data Availability

The data presented in this study are available on request from the corresponding author.

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
