# Peer review of "The Influence of Cutting Parameters on Plastic Deformation and Chip Compression during the Turning of C45 Medium Carbon Steel and 62SiMnCr4 Tool Steel"

_materials, 2022, doi:10.3390/ma15020585_

Round 1

Reviewer 1 Report

The reviewer comments of the paper «Experimental research of plastic deformation and chip compression in the cutting zone»- Reviewer

The authors presented an article «Experimental research of plastic deformation and chip compression in the cutting zone». However, there are several points in the article that require further explanation.

Comment 1:

Title needs to be concretized. Add stock material. Add machining method.

The abstract needs to be improved.

Demonstrate in the abstract novelty, practical significance. Add machining method.

Comment 2:

The introduction needs to be improved.

Firstly, group quotation is unacceptable in one phrase, for example [2-4], [5-7], [8-10], etc. Break this sentence into parts or individual sentences. For example, ... [...], ... [...], etc. Or one reference - one sentence.

Introduction, now describes the approaches of different researchers to study the process of cutting and shavings. However, in the end, the authors did not reveal what exactly is the relevance of the topic?

It is necessary to add a paragraph and a detailed analysis of the studied material of the workpiece. What difficulties are there in the machining? Why is this material so important?  What exactly has been done by scientists earlier on this material?

Now the list of references is dominated by very outdated sources, which does not add to the relevance of the work. It is unlikely that a deep analysis of such "ancient" literature is useful to the reader. It would be better if the authors replace at least 10 references published more than 10 years ago with publications from 2016-2022.

It is also helpful to add FEM analysis for similar materials. It is useful to add an orthogonal schema as well:

International Journal of Advanced Manufacturing Technology 2017, 89(9-12), 3149–3159. Doi: 10.1007/s00170-016-9216-x

Metals 2021, 11(11), 1683. doi:10.3390/met11111683

After analyzing the literature, show before formulating the goal of the "blank" spots. Which has not been previously done by other researchers. You must show the importance of the research being undertaken. Show what will be the new research approach in this article. You need to show a hypothesis.

The main thing is to clearly define for the reader what is the scientific novelty and practical significance of the article.

It is necessary to formulate a clear purpose of the article. List briefly what has been done in each section.

Comment 3:

  1. Materials and Methods

Are all figures original? If not needed appropriate citations and permissions. Refine this for figures throughout the article.

The quality and resolution of all figures needs to be improved.

Is the design of the blank in figure 1 an original decision of the authors or copied from other works?

Add the material chemistry of the TC21 stock in a separate table. What is the hardness of the workpiece and how was it measured?

Figure 5 shows the chips for which material? The quality and clarity of this figure is not acceptable and should be improved. Why is f = 0.8 used here, and 0.2, 0.4, 0.6 in Table 3?

Why are 62SiMnCr4 and C45 chosen for research? What is the hardness of these materials?

And again the question arises: what exactly is the scientific novelty and practical value of the work?

Are all formulas original? If not needed appropriate citations.

Describe the measurement procedure in more detail. At what point in time? How is the measuring setup set up? How many repetitions of measurements? What statistical methods are used to process experimental results? Describe the experimental stand in more detail. What method of experiment planning is used and why?

Comment 4:

It would be better to combine the sections and title 3. Results and discussion.

It is necessary to add a figure with the resulting chips obtained for all cutting modes investigated in the article. However, Figure 4 is not visually sufficiently understood what exactly the difference is.

The authors use well-known formulas 1 and 2. Where are the corresponding references? Where is the scientific novelty and practical significance?

Figures 6, 7, 8, 9, 10 need to be redrawn in color.

Comment 5:

Therefore, it will be useful for the authors to add, for example, the analysis of the obtained experimental data using, for example, the finite element method or other numerical methods that can do more benefits for the readers.

Or it is useful to investigate other parameters besides chips, for example, residual stresses in the surface layer, the roughness of the resulting surface.

Comment 6:

Conclusions.

It is necessary to more clearly show the novelty of the article and the advantages of the proposed method. What is the error of the obtained models? What is the difference from previous work in this area? Show practical relevance.

As it stands, the article appears to be very limited content, which is insufficient for international journals and needs to be improved. Authors should carefully study the comments and make improvements to the article step by step. After major changes can an article be considered for publication in the "Materials".

Author Response

Thank you for your comments.

Answers to Comment 1:
The title has been concretized. Stock material added. Machining method added.
The abstract has been improved.
Novelty, practical significance and machining method has been added in the abstract.

Answers to Comment 2:
The introduction has been improved.
Most group quotations have been broken into parts or individual sentences.
The relevance of the topic has been revealed in the introduction.
Detailed analysis of the studied material of the workpiece has been added.
New references have been added to the manuscript and list of references. Many old references have been removed; however, some were kept – some theoretical basics are still actual.
The “blank” spots, which has not been previously done by other researchers have been determined.
The scientific novelty and practical significance of the article have been added.
The purpose of the article has been formulated.

Answers to Comment 3:
Most of the figures are original. The citations have been added to those figures, which are not original.
Many figures have been replaced. The contrast of the figures has been increased.
The citation has been added to the schematic illustration in figure 1.
The hardness of materials has been added.
Material added to figure 5. Value f = 0.8 mm was a mistake – it was replaced by 0.6 mm.
The decision for chosen materials has been described. The hardness of the materials has been added.
The scientific novelty and practical value of the work have been added.
Citations have been added to all formulas.
The measurement procedure has been described in more detail via figure. A number of repetitions have been added.

Answers to Comment 4:
The results and discussion have been combined into a single chapter.
The figures with the resulting chips have been added.
Citations have been added to all formulas.
The different types of lines have been replaced by different colours of lines in all plots.

Answer to Comment 5:
The FEM analysis and investigation of other parameters are the priority for our further work.

Answers to Comment 6:
The novelty of the article and advantages of the obtained results have been added. The difference from previous work in this area has been added.

Reviewer 2 Report

The manuscript presents the results of a study of the cutting process of C45 and 62SiMnCr4 steels. The authors study the effect of cutting speed and feed rate on strain in the cutting zone and chip compression. The calculation of the strain is carried out according to the microstructure of the samples in the cutting zone. The results obtained are of practical interest.

Comments

  1. The introduction does not formulate the problem that the authors are solving. They wroten in the abstract: "The aim was to analyze the microstructure transverse section of cutting zone on metallographic cut." But microstructure analysis is a means, not an end. Based on the analysis of the literature, the authors must substantiate the need for their research in scientific terms and clearly set the task of the work. Without this, the work looks like a solution to a specific practical problem and is not of interest to a wide range of readers.
  2. The phrase looks completely unconvincing: “There were obtained 0.78 plastic deformation for 62SiMnCr4, however, it is considered as an error value because other values ​​are in much narrower interval. This error could be caused during improper manipulation with brittle cutting zone, or during metallography preparation procedures ". If the authors consider this result to be erroneous, then they should repeat the experiment and make sure of it, but if the result is repeated, then the effect should be recorded and tried to explain it.
  3. The authors should describe in detail the procedure for strain calculating by the microstructure of the sample. Judging by the works [Martinkovič, M.; Pokorný, P. Estimation of Local Plastic Deformation in Cutting Zone during Turning. Key Engineering Materials 662 (2015): 173 – 176; Maros Martinkovic, and Stanislav Minarik, Evaluation of Grain Deformation in Polycrystals. Materials Science Forum Vol 782 (2014) pp 41-44], it does not take into account shear strain, that is, it gives underestimated result for the effective strain. The manuscript presents the results of a study of the cutting process of C45 and 62SiMnCr4 steels. The authors study the effect of cutting speed and feed rate on strain in the cutting zone and chip compression. The calculation of the strain is carried out according to the microstructure of the samples in the cutting zone. The results obtained are of practical interest.

Author Response

Thank you for your comments.

Answer to Comment 1:
The problem that the authors are solving has been added to the introduction.

Answer to Comment 2:
The value 0.78 is the result of the calculation of the average value from three repetitions. Its high value is caused by only one number from this repetition. This value has been removed and the average value has been calculated from only two values.

Answer to Comment 3:
The procedure for strain calculating has been described in more detail via figure.

Round 2

Reviewer 1 Report

The authors have done a good job of improving the article.
However, some important comments went unheeded:
1. Introduction.
It is also helpful to add FEM analysis for similar materials. It is useful to add an orthogonal schema as well:
International Journal of Advanced Manufacturing Technology 2017, 89 (9-12), 3149-3159. Doi: 10.1007 / s00170-016-9216-x
Metals 2021, 11 (11), 1683. doi: 10.3390 / met11111683
2. The quality and resolution of all figures needs to be improved.
For example, in Figures 6 and 8, it is impossible to read the text to scale. Figure 8 needs to show the value of the angles.
3. Instead of "m.min-1" it is better to write "m / min". Check it out in all text and figures with tables.
4. In some legends, decimals are separated by "," instead of ".". For example, Figures 10-13. In addition, you need to add a, b, etc. for these and other figures and prescribe a, b, etc. in the caption below. The reader should not have to guess what the author wanted to say.

5. Are all figures original? If not needed appropriate citations and permissions. Refine this for figures throughout the article.

Author Response

Thank you for your comments.

Answer to Comment 1:
The recommended references have been added to the introduction.

Answers to Comment 2:
The quality of figures with the lower resolution has been increased. The readability of the text to scale has been increased. The value of the angles in figures 8 and 9 have been added to the figure description.

Answer to Comment 3:
The unites “m.min-1” has been rewritten to “m/min” in all text and figures with tables.

Answer to Comment 4:
The “,” for decimals has been replaced “.” in figures 10-13. It was checked in the whole manuscript. There has been added “a, b, etc.”, and they have been prescribed in mentioned figures.

Answer to Comment 5:
Figure 1a was taken from the reference [54]. It is mentioned in the text, as well as in the figure description. Rest of the figures (1b, 1c, 2, 3, 4, 5, 6, 7, 8, 9, 10, 11, 12, 13) are our original figures.

Reviewer 2 Report

Accept in present form

Author Response

Thank you very much for your approval.